# Medicarpin and Homopterocarpin Isolated from *Canavalia lineata* as Potent and Competitive Reversible Inhibitors of Human Monoamine Oxidase-B

**DOI:** 10.3390/molecules28010258

**Published:** 2022-12-28

**Authors:** Jong Min Oh, Hyun-Jae Jang, Myung-Gyun Kang, Seul-Ki Mun, Daeui Park, Su-Jin Hong, Min Ha Kim, Soo-Young Kim, Sung-Tae Yee, Hoon Kim

**Affiliations:** 1Department of Pharmacy, and Research Institute of Life Pharmaceutical Sciences, Sunchon National University, Suncheon 57922, Republic of Korea; 2Natural Product Research Center, Korea Research Institute of Bioscience and Biotechnology, Cheongju 28116, Republic of Korea; 3Department of Predictive Toxicology, Korea Institute of Toxicology, Daejeon 34114, Republic of Korea; 4National Institute of Biological Resources, Environmental Research Complex, Incheon 22689, Republic of Korea

**Keywords:** *Canavalia lineata*, medicarpin, homopterocarpin, selective human monoamine oxidase-B inhibitor, docking simulation

## Abstract

Thirteen compounds were isolated from the *Canavalia lineata* pods and their inhibitory activities against human monoamine oxidase-A (hMAO-A) and -B (hMAO-B) were evaluated. Among them, compounds **8** (medicarpin) and **13** (homopterocarpin) showed potent inhibitory activity against hMAO-B (IC_50_ = 0.45 and 0.72 µM, respectively) with selectivity index (SI) values of 44.2 and 2.07, respectively. Most of the compounds weakly inhibited MAO-A, except **9** (prunetin) and **13**. Compounds **8** and **13** were reversible competitive inhibitors against hMAO-B (K_i_ = 0.27 and 0.21 µM, respectively). Structurally, the 3-OH group at A-ring of **8** showed higher hMAO-B inhibitory activity than 3-OCH3 group at the A-ring of **13**. However, the 9-OCH3 group at B-ring of **13** showed higher hMAO-B inhibitory activity than 8,9-methylenedioxygroup at the B-ring of **12** (pterocarpin). In cytotoxicity study, **8** and **13** showed non-toxicity to the normal (MDCK) and cancer (HL-60) cells and moderate toxicity to neuroblastoma (SH-SY5Y) cell. Molecular docking simulation revealed that the binding affinities of **8** and **13** for hMAO-B (−8.7 and −7.7 kcal/mol, respectively) were higher than those for hMAO-A (−3.4 and −7.1 kcal/mol, respectively). These findings suggest that compounds **8** and **13** be considered potent reversible hMAO-B inhibitors to be used for the treatment of neurological disorders.

## 1. Introduction

Parkinson’s disease (PD) is one of the most common neurodegenerative diseases in the elderly, regardless of gender, race, or social status, and affects about 1.5 to 2.0% of people over 60 years of age and 4% of people over 80 years of age [1]. The main cause of PD is loss of dopaminergic neurons in the substantia nigra, and PD patients show major motor symptoms such as tremor and stiffness along with anxiety, depression, and dementia. Increasing dopamine (DA) level and targeting the DA-related serotonin system have been expected as anticipating areas for the treatment of PD [2]. DA levels within substantia nigra pars compacta (SNpc) neurons are maintained through synthesis of DA, synaptic vesicle loading, uptake from the extracellular space, and catabolic degradation. DA catabolism, on the other hand, begins with oxidative deamination, a reaction mediated by mitochondrial monoamine oxidase (MAO, EC 1.4.3.4) that also produces H2O2 and ammonia [3]. Therefore, increasing the concentration of DA through MAO inhibitors can be expected to relieve the symptoms by maintaining the lost DA concentration in PD patients.

Alzheimer’s disease (AD), a leading cause of dementia, is a progressive neurodegenerative disease [4]. AD affects more than 50 million people, and its main symptoms are memory and cognitive declines [5]. Various inhibitors have been being studied for the drugs of AD treatment. As one of the inhibitors, MAO inhibitors are used to treat neurological and psychiatric disorders by increasing levels of neurotransmitter such as DA, serotonin (or 5-hydroxytryptamine, 5-HT), and norepinephrine, thereby reducing MAO activity, which can cause neurological disorders [6]. On the other hand, cholinesterase (ChE) inhibitors have been studied as important tools for the treatment of AD. Acetylcholine (ACh) is a neurotransmitter in the central and peripheral nervous systems, and thus, a decrease in ACh by acetylcholinesterase (AChE) has been identified as a major cause of AD, and therefore many researches about ChE inhibitors are ongoing [7]. In addition, aggregation of beta-amyloid (Aβ) produced by beta-site amyloid precursor protein cleaving enzyme 1 (BACE1) was also studied as a cause of AD [8,9].

More specifically, MAO catalyzes the oxidative deamination of monoamines during neurotransmission with two isoforms, namely, MAO-A and MAO-B, in mitochondrial outer membranes [10]. MAO is a pharmacological target because it is involved in catecholamine and 5-HT inactivation pathways. In fact, selective reversible inhibitors of MAO-A have been targeted for anti-depression treatment, while selective reversible inhibitors of MAO-B are targets of therapeutic agents for AD and PD. Typically, selegiline, rasagiline, pargyline, and clorgyline are used as selective MAO inhibitors to reduce the symptoms of neurodegenerative and neurological diseases [11,12]. On the other hand, it was reported that MAO is also associated with the formation of amyloid plaques, which is a main cause of AD [13]. In addition, serious side effects of non-selective MAO inhibitors have been reported [14]. For this reason, selective MAO-B inhibitors based on diverse scaffolds have been actively researched (for reviews see [15,16,17]) and recently reported [18,19].

AChE (EC 3.1.1.7) shows therapeutic efficacy by increasing synaptic ACh levels in the cerebral cortex of AD patients, thereby improving cholinergic transmission [20]. As an AChE inhibitor, tacrine was used as a first drug for AD treatment. However, tacrine has been found to have a severe hepatotoxicity and is not currently used. Since then, donepezil, rivastigmine, and galantamine have been used as AChE inhibitors for therapeutic agents [11]. Like AChE, butyrylcholinesterase (BChE) is a serine hydrolase that rapidly affects the hydrolysis of the neurotransmitter ACh. The main catalytic activity of BChE is the hydrolysis of the neurotransmitters such as ACh and butyrylcholine (BCh) with a preference of BCh [21,22]. BChE is also involved in the development of the nervous system, detoxification, hydrolysis of drugs such as cocaine, heroin and aspirin, fat metabolism, and the interaction and functional modification of other proteins such as polyproline and trypsin [23]. In addition, inhibition of BACE1 related to the accumulation of Aβ for the treatment of AD patients is regarded as one of the important factors. Although many studies about BACE1 inhibitors have been reported, no one has been successful [24]. However, recently aducanumab is found to be effective to reduce Aβ plaques approved as a drug for AD by FDA [25]. Therefore, it is judged that BACE1 inhibitors are deserved to be extensively studied to reduce Aβ plaques.

Furthermore, multitargeting therapeutic strategies have been developed to target MAO-B, AChE, and/or BACE1 [26,27]. It has been reported MAO and AChE inhibitors may improve cognitive functions and alleviate symptoms in AD by elevating levels of monoamines and choline esters [28,29].

During our on-going efforts to identify potent compounds in an herbal extract library, rhamnocitrin isolated from the leaves of *Prunus padus* var. *seoulensis* was found to potently and selectively inhibit human MAO-A (hMAO-A) [30], and calycosin and 8-O-methylretusin isolated from *Maackia amurensis* were identified as selective human MAO-B (hMAO-B) inhibitors [31]. In addition, ellagic acid isolated from *Castanopsis cuspidata* var. sieboldii showed inhibition of hMAO-B and BACE1 as a multi-target inhibitor [32].

*Canavalia lineata* is a tendril perennial plant that blooms from June to August and bears fruit in October, rarely growing on the beach. *C. lineata* grows on Jeju Island in Korea and is also distributed in China, Japan, and Taiwan. It is mainly used as feed for livestock, but because of its toxicity, it was also used for abortion in the past [33]. A few physiological activities of *C. lineata* have been reported about anti-inflammation [34], and proteinase inhibition [35]. However, no study has been reported about MAO inhibitory activity of *C. lineata*. In the present study, we isolated thirteen compounds from the pods of *C. lineata* by activity-guided screening for hMAO inhibition, and evaluated their inhibitory effects on hMAO-A, hMAO-B, AChE, BChE, and BACE1, including kinetic studies and molecular docking simulation.

## 2. Results and Discussion

### 2.1. Preparation and Identification of Compounds ***1**–**13***

Constituents in *C. lineata* pod MeOH extract were separated by reverse phase column chromatography as described previously [34]. Compounds were purified by bioactivity-guided fractionation using MPLC for EtOAc and BuOH fractions and HPLC for their sub-fractions (Appendix A). Phytochemicals profile analysis of the *C. lineata* pod extract was carried out using UPLC-QTOF/MS and -PDA (Appendix A). Structures of the compounds isolated were elucidated by comparison with NMR and MS data (Appendix A) as rutin (1), (2R,3R)-3-hydroxy-7-O-β-D-glucopyranoside-6-methoxyflavanone (2), (–)-syringaresinol-4-O-β-D-glucopyranoside (3), ononin (4), (+)-syringaresinol (5), (2R,3R)-3,7-dihydroxy-6-methoxyflavanone (6), cajanin (7), medicarpin (3-hydroxy-9-methoxypterocarpan, 8), prunetin (9), 7,4′-dimethyl-3′-hydroxygenistein (10), 7,4′-dimethoxyisoflavone (11), pterocarpin (3-methoxy-8,9-methylenedioxypterocarpan, 12), and homopterocarpin (3,9-dimethoxypterocarpan, 13) (Figure 1). Compounds **6** (a flavonoid, 15.4 mg), 8 (a pterocarpan, 22.5 mg), and 13 (a pterocarpan, 31.9 mg) were detected as major metabolites of *C. lineata* pods. Compounds **8** and **13** were pterocarpans, derivatives of the isoflavonoid.

### 2.2. MAO Inhibitory Activities

A primary screening for MAO inhibition was performed using MeOH extract of *C. lineata* pods, and its EtOAc and BuOH fractions. The MeOH extract showed high MAO-B inhibition with residual activity of 53.5%. However, inhibition of MAO-A, AChE, BChE, and BACE1 inhibitions were weak (residual activities of 76.1, 78.2, 98.9, and 76.1%, respectively). After second additional extraction, MAO-B residual activity of EtOAc fraction (41.4%) was higher than that of BuOH fraction (85.9%) (Table 1). During the isolation of compounds from the extracts, thirteen compounds were identified and evaluated for their hMAO-A, hMAO-B, AChE, BChE, and BACE1 inhibitory activities. Four compounds showed MAO-A residual activities of less than 50%, and two compounds showed MAO-B residual activities of less than 10% (Table 2). In IC_50_ determination, compound **8** (medicarpin) showed the highest inhibitory activity against MAO-B with an IC_50_ value of 0.45 µM (Table 2, Appendix A), followed by **13** (homopterocarpin, IC_50_ = 0.72 µM) (Table 2, Appendix A). However, compound **12** (another pterocarpan) showed 7.47 times lower MAO-B inhibition (IC_50_ = 3.36 µM) than compound **8**. Compound **13** showed highest MAO-A inhibitory activity with an IC_50_ value of 1.49 µM, followed by **9** (prunetin, IC_50_ = 2.49 µM). Other compounds showed higher IC_50_ values for MAO-A, near 10 µM. Regarding selectivity index (SI), compounds **8** and **13** had selectivity index (SI) values of 44.2 and 2.07 for MAO-B over MAO-A, respectively (Table 2). However, most of the compounds showed low inhibitory activity against AChE, BChE, and BACE1 (Table 1).

Regarding structure–activity relationships (SAR) for hMAO-B inhibitory activities, 3-OH group at A-ring of **8** showed 1.6 times higher hMAO-B inhibitory activity than 3-OCH_3_ group at the A-ring of **13**. However, 9-OCH_3_ group at B-ring of **13** showed 4.7 times higher hMAO-B inhibitory activity than 8,9-methylenedioxy group at the B-ring of 12 (pterocarpin) (Table 2, Figure 2). Interestingly, compound **8** exhibited similar, but slightly higher (1.5 times) hMAO-B inhibition than maackiain, containing 8,9-methylenedioxy group (IC_50_ = 0.68 µM [36]) (Figure 2). When comparing **12** to maackiain, 3-OCH_3_ group instead of 3-OH group at the A-ring decreased hMAO-B inhibition but increased hMAO-A inhibition, and, additionally, 4-OH group at the A-ring of 4-hydroxy-3-methoxy-8,9-methylenedioxypterocarpan (HMMDP) [36] decreased hMAO-B inhibition as well as hMAO-inhibition.

These inhibitory activities of **8** and **13** were similar to that of an isoflavone calycosin (0.24 µM) [31], flavones acacetin (0.17 µM) [37] and acacetin 7-methyl ether (0.20 µM) [38], a pterocarpan maackiain (0.68 µM) [36], an isoflavone genistein (0.65 µM) [39]. On the other hand, **8** has been reported to have efficacy in the recovery of memory loss in scopolamine-induced mice [40], antioxidant [41], and protection of cerebral microvascular endothelial cells [42], and anti-cancer [43]. In addition, compound 13 (homopterocarpin) has been reported to have anti-cancer [44], hepatoprotective [45] and antioxidant effects [45].

### 2.3. Analysis of the Reversibility of hMAO-A and hMAO-B Inhibitions

Reversibilities of hMAO-A and hMAO-B inhibitions by compounds **8** and **13** were investigated by dialysis. Inhibition of hMAO-A by **13** was substantially recovered from 25.4% (A_U_) to 75.8% (A_D_), and these values by **13** were similar to those by the reversible inhibitor toloxatone (22.7 to 84.6%). Little recovery was observed for the irreversible inhibitor clorgyline (23.7 to 32.1%) (Figure 3). Inhibitions of hMAO-B by **8** and **13** were substantially recovered from 29.8% (A_U_) to 72.7% (A_D_), and from 32.0% to 73.5%, respectively, and these values were similar to those observed for the reversible inhibitor lazabemide (31.6 to 83.3%) (Figure 4). Besides, the irreversible inhibitor pargyline did not show activity recovery at all (34.4 to 29.1%). These results indicated compound **8** was a reversible inhibitor of hMAO-B and **13** was a reversible inhibitor of hMAO-A or hMAO-B.

### 2.4. Analysis of Inhibitory Patterns

Enzyme kinetics of compounds **8** and **13** were investigated using Lineweaver–Burk plots. Plots of hMAO-B inhibition by **8** were linear and intersected the y-axis (Figure 5A,B), indicating compound **8** is a competitive inhibitor of MAO-B. Secondary plots of the slopes of Lineweaver–Burk plots against inhibitor concentration showed the K_i_ value of 8 was 0.197 ± 0.004 µM (Figure 5B). Similarly, compound **13** was found to be a competitive inhibitor of hMAO-A or hMAO-B with K_i_ values of 0.617 ± 0.023 (Figure 6A,B) and 0.212 ± 0.008 µM (Figure 6C,D), respectively. These K_i_ values of **8** and **13** for MAO-B were slightly higher than those of the potent fluorinated chalcones **f1** and **f2** (0.027 and 0.020 µM, respectively) [46], 1-methyl, 5-phenyl substituted thiosemicarbazones **MT5** (6.58 µM) [47]. These results indicate compounds **8** was a potent competitive inhibitor of hMAO-B and compound **13** was an effective competitive inhibitor of hMAO-A and a more effective competitive inhibitor of hMAO-B.

### 2.5. Cytotoxicity Test

The effects of **8** and **13** on the viabilities of MDCK, HL-60, and SH-SY5Y cells were investigated using the CCK-8 assay. Compounds **8** and **13** showed no effects on the viabilities of MDCK (normal cell line) or HL-60 (cancer cell line) cells at 50 µM, the highest concentration tested (Figure 7A,B). However, **8** and **13** showed moderate toxicity (74.05% and 71.12% viability, respectively) to SH-SY5Y (neuroblastoma) cells at 30 µM, which is 66.7- and 41.7-times higher, respectively, than IC_50_ values of **8** and **13** (Figure 7C). These results suggest that **8** and **13** are non-toxic to the cells tested with a moderate toxicity to SH-SY5Y cells.

### 2.6. Molecular Docking Simulation

Major compounds **8** and **13**, two pterocarpans, were docked to hMAO-A and hMAO-B. Compound **12** was also included in docking simulation, because it was a pterocarpan derivative, though it was not a potent hMAO-B inhibitor. The docking simulations showed that the chemicals were located properly within the active region of hMAO-A (PDB: 2Z5X) and hMAO-B (PDB ID:4A79) (Figure 8). The compounds were docked in similar locations hMAO-A or hMAO-B, based on the surrounded amino acid residues: the compounds were located near FAD cofactor in hMAO-A, whereas they were located far from the FAD site (Figure 8). Three compounds showed higher binding affinities to hMAO-B than hMAO-A, representing they are selective hMAO-B inhibitor. Compound **8** showed the highest binding affinity (−8.7 kcal/mol) for hMAO-B as compared to other docking complexes, and formed a hydrogen bond interaction with Cys172 residue at a distance of 3.328Å (Figure 8A,B, Table 3). Compound **13** also displayed similar results, but relatively lower binding affinities for both of MAO-A (−7.1 kcal/mol) and hMAO-B (−7.7 kcal/mol) by hydrogen bond interactions with Tyr444 (hMAO-A at a distance of 3.588 Å) and Cys172 (hMAO-B at a distance of 3.333 Å), respectively (Figure 8E,F, Table 3). Compound **12**, which differs in only substituent of the B-ring of compound **13**, was predicted to have no hydrogen bond interaction with hMAO-A and hMAO-B and showed lower binding affinities (−2.4 and −7.5 kcal/mol, respectively) to both hMAO-A and hMAO-B than **8** and **13** (Figure 8C,D, Table 3). hMAO-B has relatively narrow ‘gate’ (defined by Ile199 and Tyr326) in the cavity, although hMAO-B has high sequence identity ≥70% to hMAO-A [48]. In this aspect, it might be suggested that 3-hydroxy group in **8** was more favored to hMAO-B than 3-methoxy group in **13**, and 8,9-methylenedioxy group in **12** decreased the affinity to hMAO-B compared to 9-methoxy group in **8** and **13** (Figure 8C,D). In addition, docking scores using AutoDock Vina are calculated based on considerations of various factors such as hydrogen bonding, electrostatic bonding, van der Waals forces, and dissolvent effects [49]. These scores obtained in this study were coincided well to the IC_50_ values of the compounds determined experimentally. These results suggest that compound **8** would be the most selective inhibitor against hMAO-B and compound **13** have effective inhibition activity against both hMAO-A and hMAO-B.

### 2.7. In Silico Pharmacokinetics of Medicarpin (***8***) and Homopterocarpin (***13***)

The pharmacokinetics of compounds **8** and **13** using SwissADME web tool showed these two compounds had high GI absorption and BBB permeability (Table 4). However, drug suitability of two compounds should be investigated in further study, because these compounds showed CYP450 inhibition including the important enzyme CYP3A4, although cytotoxicity to three cell lines tested above was low and some of the drugs show CYP3A4 inhibition properties [50]. Lipinski violation is judged by counting the number of violations in the Lipinski’s rule, i.e., MW < 500, cLog P_o/w_ < 5.00, HBD < 5, HBA < 10, TPSA < 140 Å^2^, and RB < 10. For Lipinski parameters of these two compounds, no violations of Lipinski’s rule of five were predicted (Table 5). These results suggest that compounds **8** and **13** can used as central nervous system (CNS) drugs.

## 3. Materials and Methods

### 3.1. General Experimental Procedures

NMR spectroscopic data were recorded by using JEOL ECZ500R [^1^H NMR at 500 MHz, ^13^C NMR at 125 MHz, (JEOL, Tokyo, Japan)] and Bruker AVANCE III HD 700 [^1^H NMR at 700 MHz, ^13^C NMR at 175 MHz, (Bruker, Billerica, MA, USA)] instruments. High-resolution electrospray ionization mass (HR-ESI-MS) data were obtained using a Vion ion mobility spectroscopy-quadrupole time-of-flight (IMS-QTOF) and ACQUITY ultra-performance liquid chromatography (UPLC) I-Class (Waters, Milford, MA, USA) system coupled with ACQUITY BEH C^18^ column (Waters). Optical rotation and CD spectra were recorded on a Jasco P-1000 polarimeter and J-815 spectrometer (Jasco, Tokyo, Japan), respectively. Medium-pressure liquid chromatography (MPLC, Spot Prep II System, Armen, Paris, France), preparative HPLC (Gilson PLC 2020 system, Gilson, Middleton, WI, USA), and multiple preparative HPLC (LC-Forte/R, YMC, Kyoto, Japan) were applied to obtain phytochemicals from *C. lineata* pod MeOH extract using C^18^ columns, YMC-Triart C18 ExRS (20.0 × 250 mm, 10 µm, YMC) and YMC ODS-AQ (20.0 × 250 mm, 10 µm, YMC).

### 3.2. Plant Materials and Preparation of Compounds ***1**–**13***

The plant materials were collected from Gujwa-eup, Jeju-do, Republic of Korea, in 2019. *C. lineata* was authenticated by National Institute of Biological Resources (NIBR, Dr. Min Ha Kim) and a voucher specimen (NIBRVP0000634114) was deposited at NIBR. In the previous study, the preparation of *C. lineata* pods extracts and the identification of phytochemicals isolated were reported by Hong et al. [34]. Briefly, the dried *C. lineata* pods (1.2 kg) were extracted with methanol (MeOH) (20 L × 3 times), then the concentrated MeOH extracts (152 g) were suspended in distilled water (1 L) and partitioned sequentially into ethyl acetate (EtOAc)- and butanol (BuOH)-soluble fractions. EtOAc and BuOH fractions were isolated by the MPLC using reverse-phase open-column (YMC ODS-AQ, 10 µm, 220 g) eluting with a gradient system of a mixture H2O and MeOH (5–100% MeOH, 50 mL/min) to yield 30 (CLPE1–CLPE30) and 25 fractions (CLPB1–CLPB25), respectively. Compounds **2** and **4**–**13** were purified from EtOAc sub-fractions (CLPE9, 14, 18, 21, 22, 24, 27, 28, and 30) using the preparative HPLC (YMC-Triart C^18^ ExRS, YMC). The preparative HPLC (YMC ODS-AQ, YMC) was used to separate compounds 1 and 3 from BuOH sub-fractions (CLPB15 and CLPB16) (Appendix A). All isolates were identified using mass fragmentation pattern and accurate mass acquired by Waters ACQUITY UPLC system equipped with QTOF mass spectrometry (Vion IMS-QTOF/MS) with ESI source and NMR spectroscopy. UPLC chromatographic condition and mass spectrometric parameters were described by Jang et al. [51].

### 3.3. Chemicals and Enzymes

Recombinant hMAO-A and hMAO-B, kynuramine, benzylamine, AChE from *Electrophorus electricus*, BChE from equine serum, 5,5′-dithiobis(2-nitrobenzoic acid) (DTNB), acetylthiocholine iodide (ATCI), butyrylthiocholine iodide (BTCI), BACE1 activity detection kit (fluorescent) and the reversible inhibitors (toloxatone, lazabemide, donepezil, quercetin) were purchased from Sigma-Aldrich (St. Louis, MO, USA) [30,31]. The reference irreversible inhibitors (clorgyline and pargyline) were obtained from Bioassay Systems (Hayward, CA, USA) [52]. All other chemicals were of reagent grade.

### 3.4. Enzyme Assays

hMAO-A and -B activities were determined using a continuous spectrophotometric method, as described previously [53,54]. The K_m_ values of hMAO-A for kynuramine and hMAO-B for benzylamine were 0.043 and 0.14 mM, respectively. The concentrations of kynuramine (0.06 mM) and benzylamine (0.3 mM) used were 1.4- and 2.1-times K_m_ values, respectively. AChE and BChE activities were assayed as described by Ellman et al. [55], with slight modification [40]. Reactions were performed using AChE and BChE in the 0.5 mL reaction mixtures including 0.5 mM substrate (ATCI and BTCI, respectively) and color reagent (DTNB). Absorbance measurements were continuously monitored for 10 min at 412 nm. Reaction rates are expressed as changes in absorbance per min. [56]. Inhibitory activities of AChE and BChE were measured after preincubating enzyme with inhibitors for 15 min before adding DTNB and each substrate. BACE1 assay was measured by the activity detection kit using a fluorescence spectrometer (FS-2, Scinco, Seoul, Republic of Korea), for 2 h at 37 °C with 7-methoxycoumarin-4-acetyl-[Asn670, Leu671]-amyloid β/A4 protein fragment 667-676-(2,4-dinitrophenyl)Lys-Arg-Arg amide trifluoroacetate as a substrate [57].

### 3.5. Kinetics of Enzyme Inhibition

Inhibitions of hMAO-A, hMAO-B, AChE, BChE, and BACE1 by the thirteen compounds were initially investigated at a concentration of 10 µM, and then IC_50_ values of the compounds and the reference inhibitors (toloxatone and clorgyline for hMAO-A, lazabemide and pargyline for hMAO-B, donepezil for AChE and BChE, and quercetin for BACE1) were determined. Kinetic parameters, inhibition types, and K_i_ values of the most potent hMAO-B inhibitors (compounds **8** and **13**) were analyzed, as described previously [31]. The kinetics of MAO-B inhibition by **8** and **13** were investigated at five different substrate concentrations (0.03–0.6 mM) and in the absence or presence of each inhibitor at three concentrations of ~0.5, 1.0, and 2.0 times of each IC_50_ value [30]. The kinetics of MAO-A inhibition by **13** was also investigated as above, except at 0.0075–0.12 mM of substrate concentrations. Inhibitory types and K_i_ values were determined using Lineweaver-Burk plots and secondary plots of their slopes, respectively.

### 3.6. Analysis of Inhibitor Reversibility

The reversibilities of hMAO-A or hMAO-B inhibitions by compounds **8** and **13** were investigated by dialysis, as previously described [53], at ~2 times of IC_50_ concentrations. Reversible and irreversible reference inhibitors were also included in the experiment at ~ 2 times of IC_50_ concentrations. After preincubating compounds or reference inhibitors with hMAO-A and hMAO-B, residual activities for undialyzed and dialyzed samples were measured. Relative values for undialyzed (A_U_) and dialyzed (A_D_) activities were then calculated and compared with non-inhibitor treated controls. Reversibilities were determined by comparing the A_U_ and A_D_ values of inhibitors with those of references.

### 3.7. Cytotoxicity

#### 3.7.1. Reagents and Cell Lines

Dulbecco’s Modified Eagle Medium (DMEM), Roswell Park memorial Institute (RPMI) 1640, Minimum Essential Medium (MEM), Fetal Bovine Serum (FBS), Penicillin/Streptomycin solution, and Trypsin-EDTA solution were purchased from Hyclone Laboratories (Logan, UT, USA). Dimethyl sulfoxide (DMSO) and 2-mercaptoethanol (2-ME) were obtained from Sigma Aldrich. Cell counting kit-8 (CCK-8) was obtained from Dojindo Laboratories (Kumamoto, Japan). Madin-Darby canine kidney (MDCK; KCLB 10034) cells, human acute promyelocytic leukemia (HL-60; KCLB 10240) cells, and human neuroblastoma cells (SH-SY5Y; KCLB 22266) were purchased from the Korean Cell Line Bank (KCLB, Seoul, Republic of Korea).

#### 3.7.2. Cell Viability

MDCK, HL-60, and SH-SY5Y cells were cultured in DMEM, RPMI 1640, or MEM, respectively [58], supplemented with heat-inactivated 10% FBS, 100 U/mL Penicillin/Streptomycin solution, and 50 µM 2-ME in a humidified atmosphere at 37 °C with 5% CO_2_ with media change every 2 days, using 75 cm^2^ T-flask, and were grown until 80% confluent [58]. Passages were used between 10 and 30. Following trypsinization, MDCK (1 × 10^4^), HL-60 (5 × 10^4^), and SH-SY5Y (5 × 10^4^) cells were seeded in 96-well plates and treated with 1, 3, 10, 30 and 50 µM of each inhibitor. The plates were then incubated for 24 h at 37 °C with 5% CO_2_, and cell viability was determined by the CCK-8 and by measuring the optical density at 450 nm using a microplate reader (Versa Max, Molecular Devices, Sunnyvale, CA, USA). The cell viability was expressed as % by comparing the absorbance of the sample to that of the control [59]. Statistical differences between groups were analyzed by ANOVA using IBM SPSS statistics 27 and probability values (*p*-values) less than 0.05 were marked as significant values.

### 3.8. Docking Simulations of the Compounds with MAO-A and MAO-B

AutoDock Vina was used to simulate docking of the chemicals to MAO-A and MAO-B [47]. The binding sites of the chemicals were selected to the binding pocket of 7-methoxy-1-methyl-9H-beta-carboline (HRM) co-crystallized with hMAO-A (PDB ID: 2Z5X), and the binding pocket of (5R)-5-{4-[2-(5-ethylpyridin-2-yl)ethoxy]benzyl}-1,3-thiazolidine-2,4-dione (P1B) co-crystallized with hMAO-B (PDB ID: 4A79). To define these binding pockets, the grid boxes for docking were centered to 40.582, 26.931, and −14.540 for hMAO-A, and 50.730, 157.601, and 30.131 for hMAO-B, with dimensions of 15 × 15 × 15 Å. The following steps were carried out to prepare the chemicals for the docking simulation: creation of 2D structures of the chemicals [60], conversion of the 2D structures into 3D structures [61], and energy minimization using the ChemOffice program (http://www.cambridgesoft.com, 23 February 2021) [61]. Docking simulations of MAO-A and MAO-B with the derivatives were performed using Chimera [62]. Based on the docking results, possible hydrogen bonding interactions were checked with relaxation constraints of 0.6 Å and 20.0° [62]. Amino acids with overlapped volume of van der Waals more than −0.4 Å with any atoms in the chemicals were defined as favorable residues.

### 3.9. Pharmacokinetic Analysis Using In Silico Method

The pharmacokinetics of the compounds were analyzed using the web tool of SwissADME (http://www.swissadme.ch/, 22 October 2022) for gastrointestinal (GI) absorption, blood–brain barrier (BBB) permeability, P-glycoprotein (P-gp) substrate, and cytochrome P450 inhibition, and skin permeation [63].

## 4. Conclusions

Thirteen compounds were isolated from the methanolic extract of *C. lineata* including EtOAc and BuOH fractions, and identified as one flavonol, two flavanones, five isoflavones, two syringaresinols, and three pterocarpans. Compounds medicarpin (**8**) and homopterocarpin (**13**) were reversible and competitive inhibitor against hMAO-B (IC_50_ = 0.45 and 0.72 µM, respectively), with high selectivity (SI = 44.2) and low selectivity (SI = 2.07), respectively. Structurally, the group 3-OH at A-ring of **8** increased hMAO-B inhibitory activity, compared to 3-OCH_3_ group at the A-ring of **13**. In addition, 9-OCH_3_ group at the B-ring of **8** and **13** showed higher MAO-B inhibition than 8,9-methylenedioxy group at the B-ring of **12**. Compounds **8** and **13** were non- or less toxic to the normal and cancer cells, including neuroblastoma cells. Molecular docking simulation revealed that **8** or **13** both bonded to Cys172 of hMAO-B with a binding affinity of −8.7 and −7.7 kcal/mol, respectively. However, compound **12** was not predicted to have hydrogen bond to MAO-B. These findings suggest that compounds **8** and **13**, a novel potent and reversible hMAO-B inhibitor, should be considered potential candidate agents for the treatment of neurological disorders.

## Figures and Tables

**Figure 1 molecules-28-00258-f001:**
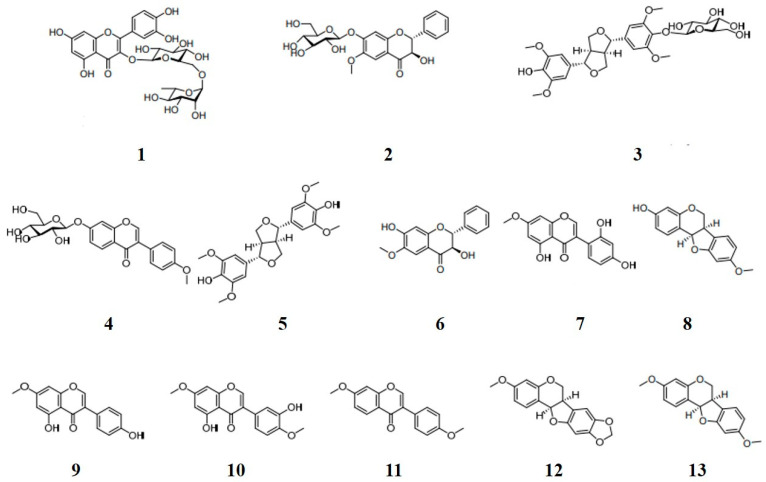
Chemical structures of compounds **1**–**13** isolated from the *Canavalia lineata*; rutin (**1**), (2*R*,3*R*)-3-hydroxy-7-*O*-d-glucopyranoside-6-methoxy-flavanone (**2**), (–)-syringaresinol-4-*O*-*β*-d-glucopyranoside (**3**), ononin (**4**), syringaresinol (**5**), (2*R*,3*R*)-3,7′-dihydroxy-6-methoxy-flavanone (**6**), cajanin (**7**), medicarpin (**8**), prunetin (**9**), 7,4′-dimethyl-3′-hydroxygenistein (**10**), 7,4′-dimethoxyisoflavone (**11**), pterocarpin (**12**), and homopterocarpin (**13**).

**Figure 2 molecules-28-00258-f002:**
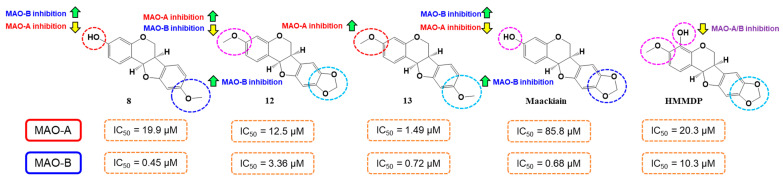
SAR analysis of pterocarpans for hMAO-A and hMAO-B inhibitions. Data of maackiain and HMMDP were from a reference [36]. HMMDP, 4-Hydroxy-3-methoxy-8,9-methylenedioxypterocarpan.

**Figure 3 molecules-28-00258-f003:**
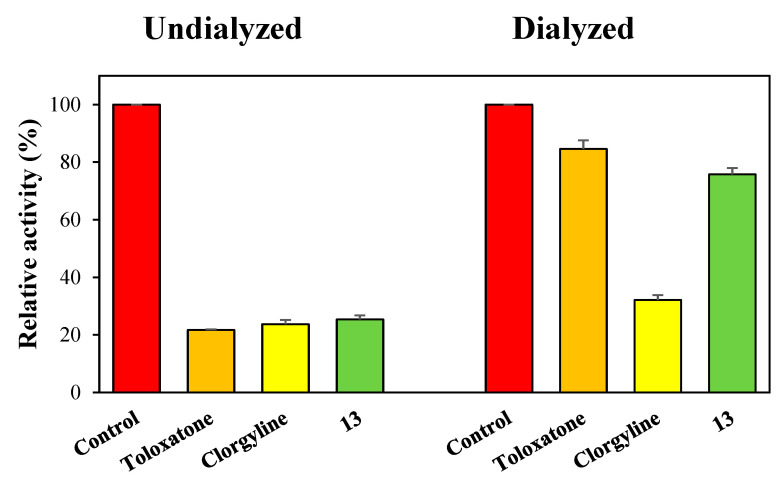
Recovery of hMAO-A inhibition by **13** after dialysis. Toloxatone and clorgyline were used as reference reversible and irreversible MAO-A inhibitors, respectively. The concentrations of the inhibitors used were as follows: **13**, 3.0 µM; toloxatone, 2.16 µM; and clorgyline, 0.014 µM. Results are the averages of experiments performed in duplicate.

**Figure 4 molecules-28-00258-f004:**
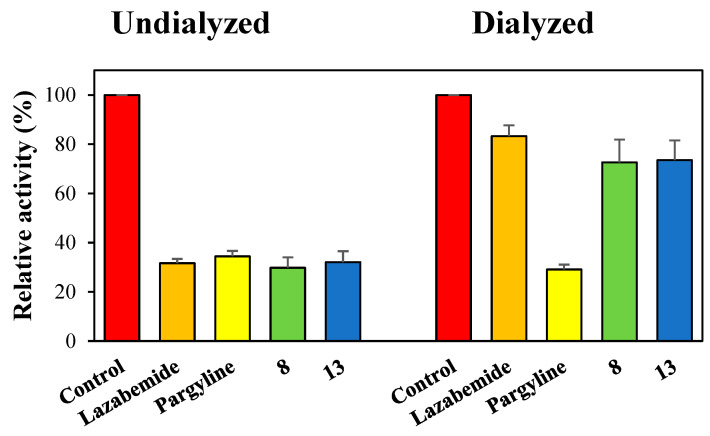
Recoveries of hMAO-B inhibitions by **8** and **13** after dialysis. Lazabemide and pargyline were used as reference reversible and irreversible MAO-B inhibitors, respectively. The concentrations of the inhibitors used were as follows: **8** and **13**, 0.90 and 1.44 µM, respectively; lazabemide, 0.22 µM; and pargyline, 0.28 µM. Results are the averages of experiments performed in duplicate.

**Figure 5 molecules-28-00258-f005:**
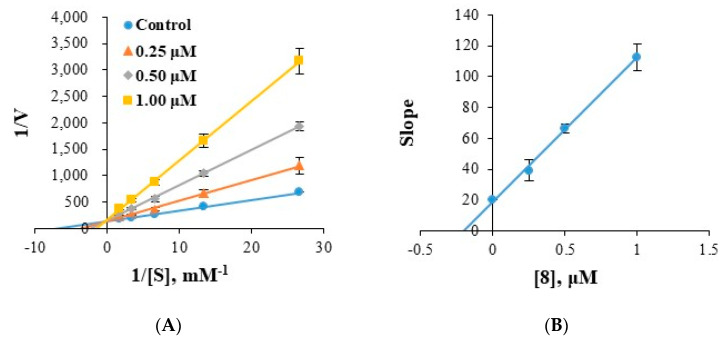
Lineweaver-Burk plots of hMAO-B inhibitions by **8** and their respective secondary plots (**A**,**B**) of slopes of Lineweaver–Burk plots versus inhibitor concentrations. Substrate concentrations ranged from 0.03 to 0.6 mM. Experiments were carried out at three inhibitor concentrations, that is, ~0.5, 1.0, and 2.0 times IC_50_ values. Initial velocity was expressed as an increase in absorbance per min.

**Figure 6 molecules-28-00258-f006:**
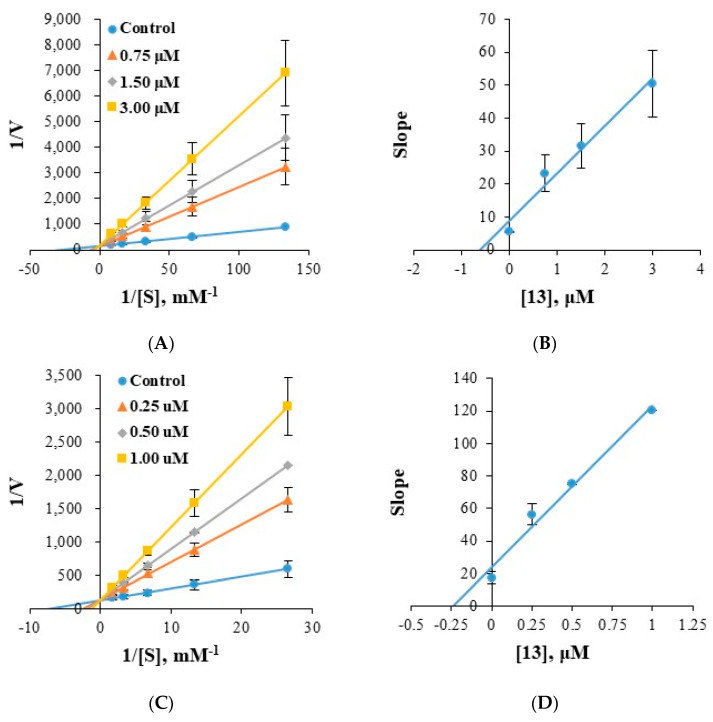
Lineweaver-Burk plots of hMAO-A and hMAO-B inhibitions by **13** (**A** and **C**, respectively) and their respective secondary plots (**B**,**D**) of slopes of Lineweaver-Burk plots versus inhibitor concentrations. Substrate hMAO-A and hMAO-B concentrations ranged from 0.0075 to 0.12, 0.03 to 0.6 mM, respectively. Experiments were carried out at three inhibitor concentrations, that is, ~0.5, 1.0, and 2.0 times IC_50_ values. Initial velocity was expressed as an increase in absorbance per min.

**Figure 7 molecules-28-00258-f007:**
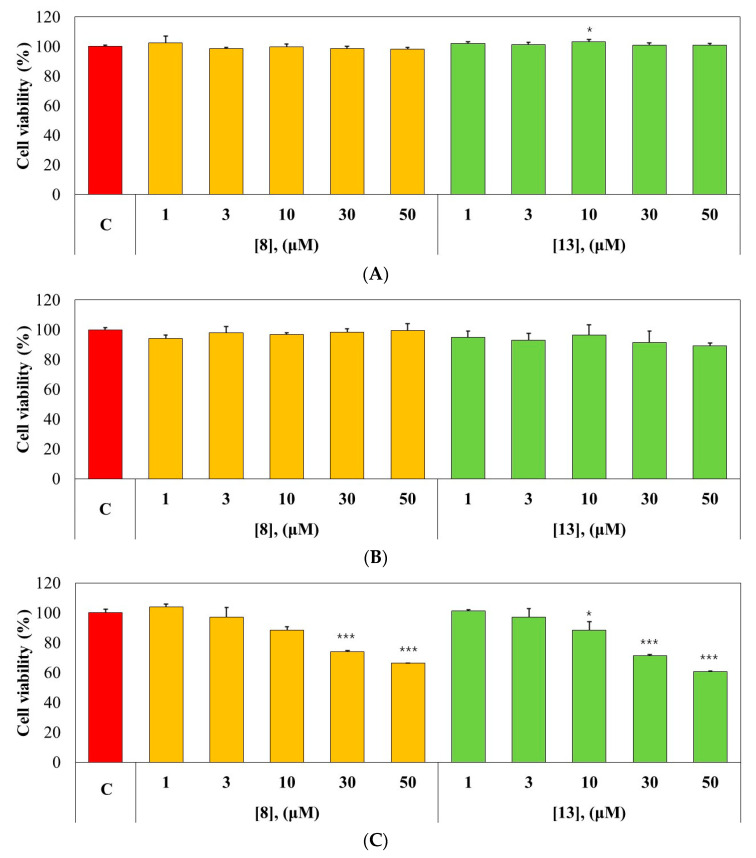
Effects of **8** and **13** on the viabilities of MDCK (**A**), HL-60 (**B**), and SH-SY5Y (**C**) cells. All cell lines were treated with the compounds (at 1, 3, 10, 30, or 50 µM) for 24 h. Culture supernatants were removed and CCK-8 was added for assay. Data are expressed as the means ± SDs of triplicate experiment. * *p* < 0.05, *** *p* < 0.001 compared to the control of each cell. C, control.

**Figure 8 molecules-28-00258-f008:**
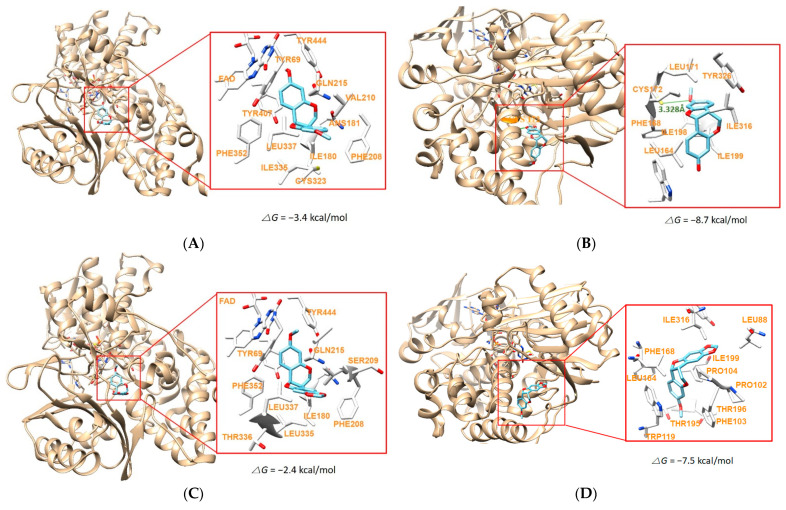
Docking simulations of **8** with hMAO-A (2Z5X) (**A**) and hMAO-B (4A79) (**B**), 12 with hMAO-A (**C**) and hMAO-B (**D**) and **13** with hMAO-A (**E**) and hMAO-B (**F**) as determined by AutoDock Vina. Green bars represent hydrogen bonds.

**Table 1 molecules-28-00258-t001:** Inhibitions of hMAO-A, hMAO-B, AChE, BChE, and BACE1 by the MeOH extract of *C. lineata* pods and its EtOAc and BuOH fractions ^a^.

Extract/Fraction	Residual Activity at 25 µg/mL (%)				
	hMAO-A	hMAO-B	AChE	BChE	BACE1
MeOH	76.1 ± 3.07	53.5 ± 8.60	78.2 ± 3.01	98.9 ± 0.50	76.1 ± 3.07
EtOAc	60.9 ± 1.54	41.4 ± 2.86	64.9 ± 0.75	99.6 ± 0.50	60.9 ± 1.54
BuOH	90.8 ± 0.77	85.9 ± 1.43	72.9 ± 5.27	97.9 ± 1.00	90.8 ± 0.77

^a^ Results are expressed as the means ± SDs of two or three experiments.

**Table 2 molecules-28-00258-t002:** Inhibitions of hMAO-A, hMAO-B, AChE, BChE, and BACE1 by the 13 compounds isolated from *C. lineata* pod extracts ^a^.

Compound	Residual Activity at 10 µM (%)					IC_50_ (µM)		SI ^b^
	hMAO-A	hMAO-B	AChE	BChE	BACE1	hMAO-A	hMAO-B	
**1**	96.79 ± 1.52	75.22 ± 6.76	92.4 ± 7.16	99.2 ± 0.88	82.56 ± 3.77	>40	>40	-
**2**	98.28 ± 2.01	69.57 ± 2.46	84.8 ± 0.00	97.5 ± 0.42	163.67 ± 2.96	>40	>40	-
**3**	80.00 ± 2.02	93.63 ± 3.47	95.22 ± 0.08	98.91 ± 1.54	76.68 ± 4.92	>40	>40	-
**4**	98.28± 1.64	63.48 ± 1.23	86.1 ± 1.79	96.7 ± 1.68	85.90 ± 0.08	>40	28.5 ± 1.55	>1.40
**5**	80.36 ± 6.57	63.24 ± 6.24	84.49 ± 0.75	99.41 ± 0.83	81.70 ± 4.01	>40	24.32 ± 2.20	>1.64
**6**	94.83 ± 1.90	94.64 ± 4.21	84.3 ± 1.31	91.8 ± 1.35	95.72 ± 0.86	>40	>40	-
**7**	56.90 ± 0.92	31.55 ± 4.21	70.8 ± 7.20	86.9 ± 2.11	83.25 ± 0.45	13.1 ± 0.23	5.18 ± 0.11	2.53
**8**	64.18 ± 0.53	4.17 ± 0.84	80.6 ± 2.62	98.4 ± 1.00	111.42 ± 0.52	19.9 ± 0.85	0.45 ± 0.032	44.2
**9**	24.75 ± 1.22	30.81 ± 0.82	83.2 ± 4.76	93.0 ± 0.65	87.73 ± 0.59	2.49 ± 0.32	5.18 ± 0.059	0.48
**10**	41.02 ± 1.09	16.86 ± 2.47	78.4 ± 3.40	98.1 ± 1.98	78.86 ± 0.56	8.05 ± 0.10	3.47 ± 0.055	2.32
**11**	49.15 ± 2.25	62.02 ± 1.09	82.7 ± 5.44	98.7 ± 0.66	91.96 ± 0.59	9.80 ± 0.27	13.7 ± 0.10	0.72
**12**	60.34 ± 0.90	13.01 ± 0.72	84.9 ± 0.61	93.6 ± 2.11	102.36 ± 0.40	12.5 ± 0.33	3.36 ± 0.090	3.72
**13**	11.59 ± 2.60	4.05 ± 1.91	81.0 ± 2.44	88.5 ± 1.96	101.48 ± 0.01	1.49 ± 0.021	0.72 ± 0.028	2.07
Toloxatone			-	-	-	1.080 ± 0.025	-	
Lazabemide			-	-	-	-	0.110 ± 0.016	
Clorgyline			-	-	-	0.007 ± 0.001	-	
Pargyline			-	-	-	-	0.140 ± 0.006	
Tacrine *			0.270 ± 0.019	0.060 ± 0.0022	-	-	-	
Donepezil *			0.010 ± 0.002	0.180 ± 0.0038	-	-	-	
Quercetin *			-	-	13.420 ± 0.035	-	-	
Inhibitor IV *					0.440 ± 0.064			

^a^ Results are expressed as the means ± SDs of two or three experiments. ^b^ SI values for hMAO-B selectivity were calculated by dividing IC_50_ values of hMAO-A by those of hMAO-B. * Values were represented as IC_50_ values of reference compounds against AChE, BChE, and BACE1.

**Table 3 molecules-28-00258-t003:** Docking scores and predicted hydrogen bond(s) of the three compounds with hMAO-A or hMAO-B *.

Compound	Docking Energy (kcal/mol)		H-Bond Predicted	
	hMAO-A	hMAO-B	hMAO-A	hMAO-B
**8**	−3.4	−8.7		Cys172 (3.328 Å)
**12**	−2.4	−7.5		
**13**	−7.1	−7.7	Tyr444 (3.588 Å)	Cys172 (3.333 Å)

* Determined by AutoDock Vina.

**Table 4 molecules-28-00258-t004:** Pharmacokinetic properties of medicarpin and homopterocarpin.

Compound	GI Absorption	BBB Permeant	P-gp Substrate	Inhibitor					Log K_p_
				CYP1A2	CYP2C19	CYP2C9	CYP2D6	CYP3A4	Skin Permeation (cm/s)
**8**	High	Y	Y	Y	Y	N	Y	Y	−5.98
**13**	High	Y	Y	Y	Y	N	Y	Y	−5.84

GI, gastrointestinal; BBB, blood–brain barrier; P-gp, P-glycoprotein.

**Table 5 molecules-28-00258-t005:** Physicochemical parameters and Lipinski violations.

Compound	Mw (g/mol)	cLog *P*	HBD	HBA	TPSA (Å^2^)	RB	Lipinski Violations
**8**	270.28	2.53	1	4	47.92	1	0
**13**	284.31	2.91	0	4	36.92	2	0

Mw, molecular weight; cLog *P*, consensus Log *P*, HBD: H-bond donors; HBA, H-bond acceptors; TPSA, topological polar surface area; RB, rotatable bonds.

## Data Availability

Not applicable.

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
