# Peer review of "Medicarpin and Homopterocarpin Isolated from Canavalia lineata as Potent and Competitive Reversible Inhibitors of Human Monoamine Oxidase-B"

_molecules, 2022, doi:10.3390/molecules28010258_

Round 1

Author Response

Response to Reviewers’ comments

Reviewer 1

Introduction

  • There is need to sequentially compile data in introduction.
  • Thank you for your comments. Introduction was revised according to target enzymes.
  • The sentence at L55 “Typically, selegiline, pargyline, and clorgyline are used as MAO inhibitors to reduce the symptoms” was moved to L67 in the revision by revising to “Typically, selegiline, rasagiline, pargyline, and clorgyline are used as selective MAO inhibitors to reduce the symptoms of neurodegenerative and neurological diseases.“ In addition, description of BChE at L62 “Like AChE, butyrylcholinesterase (BChE) is a serine hydrolase that rapidly affects the hydrolysis of the neurotransmitter ACh. The main catalytic activity of BChE is the hydrolysis of the neurotransmitters such as ACh and butyrylcholine (BCh) with a preference of BCh [11,12]. BChE is also involved in the development of the nervous system, detoxification, hydrolysis of drugs such as cocaine, heroin and aspirin, fat metabolism, and the interaction and functional modification of other proteins such as polyproline and trypsin [13].” was also moved to behind the AChE introduction at L80. The reference numbers were changed relevantly.

  • How MOA inhibitors regulate the level of dopamine. Revise line no 53-54.
  • Thank you for your careful comments. The word “reducing” in front of “levels of neurotransmitters” was corrected to “increasing”.
  •  
  • Need to add and discuss previously reported inhibitors of MAO.
  • Thank you for your comments. The sentence at L55 was revised and moved to L67 in this revision, as “Typically, selegiline, rasagiline, pargyline, and clorgyline are used as selective MAO inhibitors to reduce the symptoms of neurodegenerative and neurological diseases [11,12].”.

Results and discussion

  • Correct the caption of Table. 4 (Caption of Table. 2 was copied here).
  • Thank you for your careful review. We corrected.

 Physiochemical properties of selective compounds were not discussed under the heading 2.6

  • Thank you for your comments. The sentence was added at L306: “Lipinski violation is judged by counting the number of violations in the Lipinski’s rule, i.e., MW < 500, cLog Po/w < 5.00, HBD < 5, HBA < 10, TPSA < 140 Å2, and RB < 10.”

Reference and citation 

  • Add reference correctly in line no 170
  • Thank you for your comments. We changed.

  • Revise reference no 60 in line 432
  • Thank you for your comments. We changed.

Thank you for your valuable comments.

Reviewer 2 Report

Point 1:

The study predicted that 5 rings could prohibit the interaction with hMAO-B binding compared to 4 rings in (8), (13) with the 3-hydroxyl substituent that couldbind more efficiently than 3-methoxyl to MAO-B (hydrogen bond interaction with Cys172) so (12) couldnot bind with MAO-B as good as (8) and (13).However, IC50 value of Maackian structure has 5 rings with similar 3-hydroxyl substituent same with (12) hasan effective IC50 value with MAO-B.

Point 2: 

The authors should mention the binding site in docking stage. What is the binding site and how did the study identify important acid amines. 

Point 3: 

According to pharmacokinetic properties of medicarpin and homopterocarpinthese substances might beinhibited by CYP450, including the important enzyme CYP3A4. Therefore, will these substances suitable to be used a potential drug? Will they have the high risk of drug interaction? The authors should further discuss.

Reviewer 3 Report

In general, the topic of this paper is actual and important for the improvement of treatment of  neurological disorders such as Parkinsonʼs and Alzheimer`s diseases. The paper describes isolation of thirteen compounds from the methanolic extract of Canavalia lineata pods. After their identification, their inhibitory activity against hMAO-A, hMAO-B, AChE, BChE and BACE1 was evaluated. Based on the results, two compounds, medicarpin (compound 8) and homopterocarpin (compound 13) seem to be effective reversible competitive inhibitors of hMAO-B. Both compounds were shown to be selective inhibitors of hMAO-B, nevertheless, in my opinion, only compound 8 is a real selective inhibitor of hMAO-B while compound 13 is also very effective inhibitor of hMAO-A and its selectivity is much lower compared to the compound 8 (selectivity index values for hMAO-B over hMAO-A were 2.07 for compound 13 and 44.2 for compound 8). Both compouds are non-toxic to the normal and cancer cells (MDCK and HL-60 cells) but moderately toxic to human neuroblastoma cells (SH-SYSY cells). The docking simulations showed that their binding affinities for hMAO-A and hMAO-B correspond to their inhibitory activities. According to in silico pharmacokinetic study, both compounds have high gastrointestinal absorption and blood-brain barrier permeablility. It means that both compounds can be used as central nervous system drugs. Based on the obtained results, both compounds can be considered to be promising agents for the treatment of neurological disorders. The experimental methods used in this study are modern and adequate to the investigated topic. The results are clearly demonstrated and sufficiently discussed. Nevertheless, there are some shortcomings in this paper.

Shortcomings

·         Introduction, page 3, line 109-110 - the sentence „and observed that medicarpin and homopterocarpin potently and selectively inhibited hMAO-B“ is not suitable for Introduction because it brings information about the results of this study. Therefore, it should be omitted.

·         Results and Discussion, page 10, line 256 – the subtitle „2.5 Cytotoxicity test“ is wrong. The correct subtitle should be  2.6. Docking simulations

·         Results and Discussion, page 10 line 261 and 264 – figure 7 gives data about cytotoxicity of studied compounds, not about docking simulations. Thus, „figure 8“ should be correct.

·         Results and Discussion – page 12, line 304 – „2.7“ instead of „2.6“. 

·         Conclusionthe compound 13 (homopterocarpin) was demonstrated to be selective inhibitor of hMAO-B but this fact is not clear. The compound 13 is very effective inhibitor of both isoforms of hMAO (A,B) and its selectivity is low (the selectivity index value was only 2.07 while the selectivity index value for the compound 8 was 44.2). In addition, the binding affinities of the compound 13 for hMAO-A and hMAO-B are similar. Therefore, only medicarpin seems to be a real selective inhibitor of hMAO-B. The authors should explain why they characterize the compound 13 (homopterocarpin) as selective inhibitor of hMAO-B.

Round 2

Reviewer 2 Report

All the concern comments are solved. 

Author Response

Thank you for your kind review.